# In-Shoe Sensor Measures of Loading Asymmetry during Gait as a Predictor of Frailty Development in Community-Dwelling Older Adults

**DOI:** 10.3390/s24155054

**Published:** 2024-08-04

**Authors:** Tatsuya Nakanowatari, Masayuki Hoshi, Akihiko Asao, Toshimasa Sone, Naoto Kamide, Miki Sakamoto, Yoshitaka Shiba

**Affiliations:** 1Department of Physical Therapy, Fukushima Medical University School of Health Sciences, 10-6 Sakae-machi, Fukushima 960-8516, Fukushima, Japan; 2Department of Occupational Therapy, Fukushima Medical University School of Health Sciences, 10-6 Sakae-machi, Fukushima 960-8516, Fukushima, Japan; 3School of Allied Health Sciences, Kitasato University, 1-15-1 Kitasato, Minami-ku, Sagamihara 252-0373, Kanagawa, Japan

**Keywords:** frailty, gait, wearable sensor, plantar force, asymmetry

## Abstract

Clinical walk tests may not predict the development of frailty in healthy older adults. With advancements in wearable technology, it may be possible to predict the development of frailty using loading asymmetry parameters during clinical walk tests. This prospective cohort study aimed to test the hypothesis that increased limb loading asymmetry predicts frailty risk in community-living older adults. Sixty-three independently ambulant community-living adults aged ≥ 65 years were recruited, and forty-seven subjects completed the ten-month follow-up after baseline. Loading asymmetry index of net and regional (forefoot, midfoot, and rearfoot) plantar forces were collected using force sensing insoles during a 10 m walk test with their maximum speed. Development of frailty was defined if the participant progressed from baseline at least one grading group of frailty at the follow-up period using the Kihon Checklist. Fourteen subjects developed frailty during the follow-up period. Increased risk of frailty was associated with each 1% increase in loading asymmetry of net impulse (Odds ratio 1.153, 95%CI 1.001 to 1.329). Net impulse asymmetry significantly correlated with asymmetry of peak force in midfoot force. These results indicate the feasibility of measuring plantar forces of gait during clinical walking tests and underscore the potential of using load asymmetry as a tool to augment frailty risk assessment in community-dwelling older adults.

## 1. Introduction

The global prevalence of frailty is over 17%, and it increases with age [1]. Frailty is a weakness syndrome associated with aging, representing a physiological decline closely related to adverse events such as falls, disorders, and death [2,3]. Most frailty-specific interventions do not target older adults before the onset of frailty. Preventive interventions are likely to have a greater impact on the overall health of the population [4].

One of the challenges in developing primary preventions for frailty is the difficulty in identifying high-risk individuals among older adults without clear signs of poor health. Currently, technological advances provide more accurate gait analysis methods to detect frailty risk in older people. In particular, wearable sensors offer the advantage of monitoring subjects anywhere without needing a controlled environment. Several studies [5,6,7,8] have supported the use of gait analysis systems for the evaluation of frailty risk in older people.

Ground reaction force is commonly used in gait analysis to distinguish between healthy and pathologic subjects and plays a crucial role in identifying biomechanical factors of disease progression in older adults [9,10]. Specifically, ground force has been used as the primary metric in a biofeedback training protocol and as both an assessment tool and a secondary metric in older adults [11]. Instead of expensive equipment to measure ground reaction forces, load-sensing insoles enable the collection of force-based metrics through wearable technology, enhancing the accessibility of gait assessments. The new triple-sensor load measuring insoles (loadsol^®^, Novel Electronics, Munich, Germany) provide solutions to the difficulties of gait mechanics monitoring. They work with capacitive sensors, feature a wireless design using Bluetooth communication, and are capable of measuring the force applied to the ground by the plantar overall region, as well as three sub-regions: forefoot, midfoot, and rearfoot. A capacitive sensor consists of a compressible, soft dielectric material in the middle and two electrode layers. If a force or pressure is applied to a capacitive sensor in the vertical direction, the distance between the electrode layers is reduced, which changes the capacity of the sensor. This change in the alternating current through the sensor is subsequently evaluated by the measuring electronics connected to the sensor. The loadsol^®^ requires the use of an app-enabled mobile device, rendering it a portable and clinically accessible option for collecting force-based data [12,13,14,15]. It has been shown to be a valid and reliable method of collecting load kinetics in older adults [16]. Additionally, with the use of loadsol^®^, we previously found increased loading asymmetry on the plantar region during walking in community-living older adults with a history of falls [17]. This finding indicates the importance of assessing loading asymmetry to screen frailty risk in older adults because frailty is closely related to fall risk. If asymmetry in loading, as measured by the loadsol^®^, proves to be a reliable predictor of frailty risk, this would suggest that the loadsol^®^ is effective in detecting unnoticed problems of locomotion through its collection of load kinetics. Moreover, if a regional plantar force that contributes to loading asymmetry on net forces could be identified, it would help to establish a novel metric of gait training in biofeedback to prevent frailty.

The primary objectives of this investigation were, therefore, (1) to test the hypothesis that increased limb loading asymmetry on net plantar force during walking predicts the development of frailty in community-dwelling older adults participating in a community health survey and (2) to examine the relationship of loading asymmetry between net and regional plantar forces in this population. For these objectives, we propose to address the following contributions.

Firstly, we evaluate differences between older adults who developed frailty and those who did not use clinical walk tests and in-shoe sensor measures. We aim to explore the association between asymmetry parameters of gait and frailty development in older adults and identify possible key parameters to classify individuals according to device parameters.

Secondly, we analyze whether key parameters predict frailty development and determine which sub-region asymmetry (forefoot, midfoot, or rearfoot) is correlated with key parameters. Our aim is to examine the utility of asymmetry parameters of gait as a predictor of frailty development and to elucidate mechanisms contributing to the asymmetry of gait and risk of frailty development.

## 2. Materials and Methods

### 2.1. Participants

We conducted a prospective cohort study through a community health survey in Minamisoma City, Fukushima Prefecture, Japan. The participants were recruited via advertisements in community newsletters. To be eligible to take part in the present study, the participants had to be 65 years or older, community-dwelling, and able to be independent in activities of daily living (ADL). The exclusion criteria included meeting frailty criteria at baseline, the use of walking aids, and limitations preventing them from participating in the gait and physical function tests described below.

Ethical approval for this study was granted by the Institutional Review Board of Fukushima Medical University (reference number 2022-123). The participants were provided with an information sheet explaining the study.

### 2.2. Procedures

After informed consent was obtained, the subjects filled out a questionnaire and performed the gait and physical function tests at baseline. Ten months after this baseline assessment, a follow-up assessment was conducted to obtain information regarding frailty. In this study, loading asymmetry during gait was evaluated for frailty development prediction by using datasets in the community-based cohort (Figure 1).

### 2.3. Assessment of Loading Asymmetry

Each participant was fitted with a pair of standardized laboratory-issued walking shoes (Taketora Inc., Kanagawa, Japan), with the loadsol^®^ placed within the shoes (Figure 2). The force sensors are segmented by the manufacturer into thirds based on sensor length, representing rearfoot, midfoot, and forefoot plantar regions. The force sensors were calibrated using a previously described protocol [13,14]. Participants were instructed to load the insole with their full body weight in a single peg stance, then unload the insole, and again to load the insole on each foot. The loading of the participant was entered in Newton (N) in the loadsol^®^ application on an iPad mini (Apple Inc., Cupertino, CA, USA) and was recorded at 100 Hz. 

Following calibration, each participant performed the 10-meter walk test (10MWT) twice. Participants were instructed to walk as quickly but as safely as possible. First, they walked along a 14 m straight walkway on a flat, hard surface floor at their maximum speed. Then, they slowly turned around and walked back along the 14 m straight walkway at their maximum speed. The turning phase was used to segment the walking data.

Data were processed using loadpad analysis software (Novel Electronics). The raw plantar force data proceeded without a Butterworth filter using a cutoff based on a previously reported analysis [14]. The calibrated plantar forces were imported as a net value and three regional values. Each participant’s steps were identified within walking using a 30 N threshold (≥6% of mean participant body weight) from the net force–time-series data to avoid false positive step identification when the force sensor deflected within the shoe. Plantar forces data were extracted for the middle 10 steps within a straight walking phase and calculated to four variables of interest (contact time [CT], peak force [PF], loading rate [Lr], and impulse) of each left/right for each trial (Figure 3). CT was defined as the time in which the total force signal of the insole is equal to or higher than 30 N. PF was defined as the maximum force value of each step in the period of 0–40% of the stance phase. Lr was defined as the slope of 20% to 80% into the first peak of the force curve. Impulse was defined as the force–time integral between the force curve and the time axis. CT, PF, and impulse were calculated for net, rearfoot, midfoot, and forefoot regions, and Lr was calculated for net region only. For left and right steps independently, CT, PF, and Lr were averaged over five steps, and impulse was summed with those of five steps. Finally, all variables were averaged over two trials, and PF, LR, and impulse were normalized to body weight.

Additionally, asymmetry parameters for each of all variables were calculated by the difference between the more loaded foot and the less loaded foot and then divided by the sum of both feet [18]. CT, PF, Lr, and impulse asymmetry parameters were calculated using the following formula:
CT asymmetry (%) = abs (Mean *CT*
_Right_ − Mean *CT*
_Left_)× 100Mean *CT*
_Right_ + Mean *CT*
_Left_PF asymmetry (%) = abs (Mean *PF*
_Right_ − Mean *PF*
_Left_)× 100Mean *PF*
_Right_ + Mean *PF*
_Left_Lr asymmetry (%) = abs (Mean *Lr*
_Right_ − Mean *Lr*
_Left_)× 100Mean *Lr*
_Right_ + Mean *Lr*
_Left_Impulse asymmetry (%) = abs (Mean *Impulse*
_Right_ − Mean *Impulse*
_Left_)× 100Mean *Impulse*
_Right_ + Mean *Impulse*
_Left_

### 2.4. Evaluation of Frailty

The Kihon Checklist (KCL), which allows comprehensive assessment of frailty in daily life [19,20], was used to assess frailty at baseline and follow-up. The KCL comprises 25 items (yes/no questions) that assess important areas of frailty, including ADL (Items 1–5), physical function (Items 6–10), nutritional status (Items 16 and 17), cognitive function Items 18–20) and depressive mood (Items 21–25). The participants were classified into three groups based on the KCL. Out of a maximum of 25 points, those with scores of ≥8 points were defined as the frailty grading, those with 4–7 points were classified as the pre-frailty grading, and those with ≤3 points as the robust grading [21]. The frailty development was defined if the participant progressed from baseline at least one grading group of frailty at the follow-up period. The frailty not-development was defined if the participants sustained or improved their frailty grading at the follow-up period from baseline.

### 2.5. Assessment of Covariates

The gait speed, the Timed Up and Go (TUG) test, fall history, presence of comorbidities, leg pain, and number of medications were collected as confounding factors at baseline. The straight walking trials at their comfortable speed were timed in the middle 10 m, between marks 2 and 12 m, and its data were expressed as speed (m/s) [22]. The TUG test was performed to rise from an armchair, walk 3.0 m at the participant’s maximum speed to a mark, turn around, return to the chair, and sit down [23]. History of falls in the past year and presence of cerebrovascular disorder, hypertension, heart disease, diabetes, liver disease, kidney disease, and lung disease or/and asthma were retrospectively investigated by questionnaire. In the past year, those who reported one or more falls were defined as fallers, and those who reported no falls were defined as non-fallers. Knee and/or foot pain was assessed using a questionnaire asking whether the participants had experienced pain lasting for more than one month.

### 2.6. Statistical Analysis

To determine potential confounding factors, an independent *t*-test for continuous variables and *ꭓ*^2^ test for categorical variables at baseline were used to test for statistical differences between participants who developed frailty and those who did not. Analysis of covariance (ANCOVA) adjusted for potential confounding factors was performed, with the occurrence of frailty development set as the independent variable and the net asymmetry parameters set as the dependent variables. Following convention [24], multivariate logistic regression was used to predict net asymmetry parameters associated with frailty development. Age, gender, gait speed, and KCL score were included as covariates in the model. Additionally, correlations between the predictive values of net asymmetry and rearfoot, midfoot, and forefoot asymmetry were tested by using Pearson’s correlation coefficient. A 2-sided *p*-value < 0.05 was considered statistically significant. Statistical analysis was performed using SPSS software (Version 29, SPSS Inc., Chicago, IL, USA).

## 3. Results

### 3.1. Subject Characteristics

Sixty-three subjects participated in the baseline of this study. None of them used walking aids. Of the participants, nine subjects did not attend the 10-month follow-up visit. Six subjects who met the frailty criteria at baseline and one subjects with missing force loading data were excluded (Figure 4). There were no significant differences regarding baseline characteristics, performance, and loading asymmetry parameters in the subjects with and without follow-up. The mean age of the remaining 47 subjects (37 women, 10 men) was 75.3 ± 5.8 years. The mean height and weight were 153 ± 7 cm and 53.2 ± 9.5 kg, respectively. Gait and balance functions were relatively high; the mean comfortable gait speed was 1.45 ± 0.18 m/s, and TUG time was 6.4 ± 1.1 s.

According to the KCL, the prevalence of robustness and pre-frailty at baseline were 62% (*n* = 29) and 38% (*n* = 18), respectively. At the follow-up period, the 14 participants were defined as the frailty development group and the 33 participants as the frailty non-development group (Figure 5). The baseline variables of the frailty development and non-development groups are shown in Table 1. The KCL scores at baseline, that the frailty development group had a significantly lower score than the non-development group (*p* = 0.03), was used as a covariate for ANCOVA.

### 3.2. Loading Asymmetry and Frailty Development

The ANCOVA analyses, run with the KCL scores as a baseline as a covariate, showed that two values of loading asymmetry on net plantar force were significantly increased in those who developed frailty (*n* = 14) compared with those who did not (*n* = 33), as summarized in Figure 6. Net Lr asymmetry in the frailty development group was significantly higher than that in the frailty non-development group (F = 4.432, *p* = 0.042). Also, Net impulse asymmetry in the frailty development group was significantly higher than that in the frailty non-development group (F = 9.647, *p* = 0.004).

Multivariate logistic regression analysis showed that the risk factor independently associated with frailty development included the net impulse asymmetry (OR = 1.269, 95%CI 1.016–1.585, *p* = 0.033) and the KCL scores (OR = 2.331, 95%CI 1.170–4.647, *p* = 0.016) (Table 2). The net Impulse asymmetry was associated only with PF asymmetry on midfoot (*r* = 0.323, *p* < 0.05). Other loading asymmetry variables on the regional plantar forces did not correlate with net impulse asymmetry (Table 3).

## 4. Discussion

The present study of loading asymmetry in gait has several key findings. First, among community-dwelling older adults without frailty, net impulse asymmetry in gait is a valid predictor of frailty development within ten months. Second, net impulse asymmetry in gait is associated with PF asymmetry in the midfoot region, where the medial longitudinal arch of the foot plays an important role in shock attenuation and generating sufficient power for propulsion during gait. The concomitant study of kinetics and clinical data may also help to elucidate further the mechanisms that contribute to gait asymmetry and the risk of frailty development. Our findings suggest that the use of in-shoe sensors may address the challenges of their implementation in clinical settings, and loading asymmetry during gait may be useful as an indicator of frailty development risk.

The present findings are consistent with previous results obtained in slightly different settings and populations. First, Pinloche et al. [25], using plantar sensors for foot pressure, reported that COP (center of pressure) velocity and projection differed between frailty and pre-frailty groups in a nursing home population. Second, Anzai et al. [26], using smart insoles for plantar pressure, found that classifying participants relative to their frailty state based on the KCL relied on features extracted from the plantar pressure series during walking. Additionally, regarding the frailty condition at baseline, Fried et al. [2] reported that the pre-frailty stage is identified as a high risk of progressing to frailty. The present study extends these findings in several ways. Our results indicate the feasibility of obtaining measures of loading asymmetry using a single, easy-to-use instrument with a clinical walk test. Furthermore, the present study shows the relationship between loading asymmetry in gait and the future development of frailty in community-dwelling older adults. This suggests that the in-shoe sensor device can be used to identify older adults at risk of frailty development, allowing clinicians to intervene early to prevent the onset of frailty.

Increased net impulse asymmetry was associated with increased midfoot PF asymmetry. Given that the midfoot, with its medial longitudinal arch, is widely recognized as the shock attenuation and the propulsive part of the foot, it could be hypothesized that these midfoot functions may decline in people at frailty risk. Age-related changes in posture, including accentuated plantar deformation, make older adults more cautious, which reduces walking symmetry [27,28]. Some explanations could be given, such as tibialis posterior dysfunction [29], decreased strength, sensitivity or mobility of the foot, or alteration of the somatosensory system. The observed midfoot changes suggest that older adults may adopt a ‘pull-off’ rather than ‘push-off’ strategy to generate forward momentum during walking. This reliance on various musculoskeletal and biomechanical factors may help explain the predictive value of loading asymmetry in gait.

The present study has several limitations. First, the relatively small sample size may have resulted in a Type II error or a failure to detect other differences between the developed and non-developed frailty groups. Other measures of loading asymmetry might have also been expected to differ significantly between the developed and non-developed frailty groups. Increasing the sample size would reduce the risk of Type II errors and allow further detailed studies of (1) the relationship between net and regional plantar forces in relation to load asymmetry or (2) the specificity and sensitivity of various predictors of frailty risk. Second, a larger sample size might have enabled us to assess the generalizability of these findings. Future studies with a larger sample population might help elucidate the association between the risk frailty development and loading asymmetry of gait. Third, the participants in the study included individuals who were in a pre-frailty condition. To increase the internal validity of our findings, we could consider studying only older adults with robust conditions or comparing older adults with young adults.

We should also consider additional limitations related to the standards adopted in this study to identify frailty. Although the KCL is widely used in Japan and has been validated as a frailty prediction tool in several reports [21,30], many other criteria are available to predict frailty. Ambagtsheer et al. [31] suggested that results could vary widely, especially between self-administrated methods, such as the Kihon checklist, and tests administered by nurses or physicians. It is possible that the accuracy of our study could have been different, either increased or decreased, if another frailty assessment tool had been used as a reference instead of the KCL.

Finally, another limitation of the present study is the non-inclusion of variables related to other diseases that cause joint pain, such as arthritis. A combination of gait loading asymmetry and such diseases could potentially improve the accuracy of frailty condition prediction. Future studies are needed to investigate the impact of asymmetry due to a history of joint pain using questionnaires and physical examinations.

## 5. Conclusions

Limb loading asymmetry, measured with a 3-sensor insole during a 10 m walking test, has proven effective in identifying older adults at risk of frailty. Our results suggest that assessing limb loading asymmetry in gait could help in predicting frailty development in older adults through community health screening. Although further research is needed to fully understand the various factors contributing to gait asymmetry and its relationship to frailty risk, our findings may be valuable for implementing new or existing interventions that use limb loading as biofeedback to prevent frailty.

## Figures and Tables

**Figure 1 sensors-24-05054-f001:**
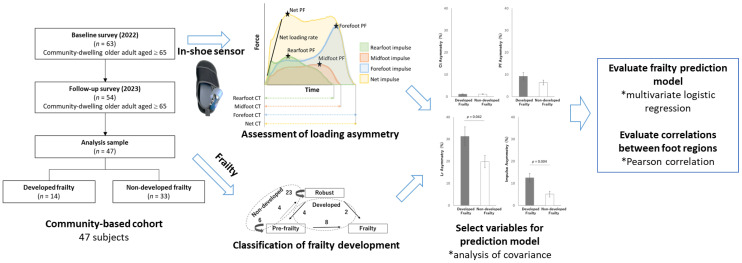
Study workflow.

**Figure 2 sensors-24-05054-f002:**
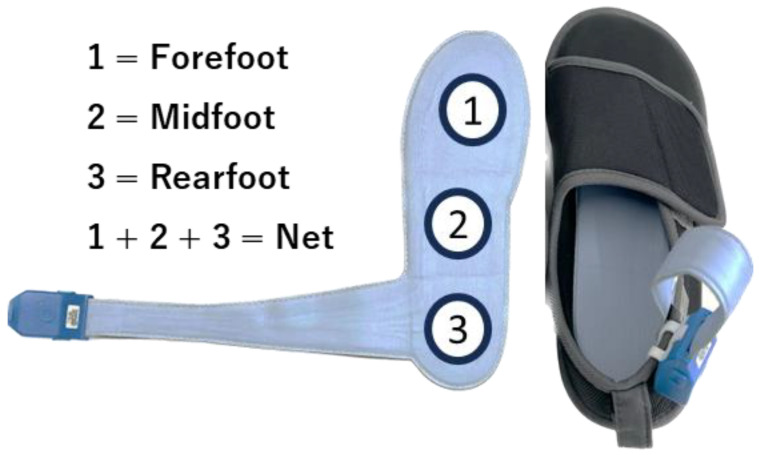
Overview of the three-sensor plantar force measurement insole device. Three sensors are inserted into a 3-mm-thick shoe insole. The insole is inserted in a commercial Velcro shoe, and the data acquisition unit is fixed using an attached clip on the shoe.

**Figure 3 sensors-24-05054-f003:**
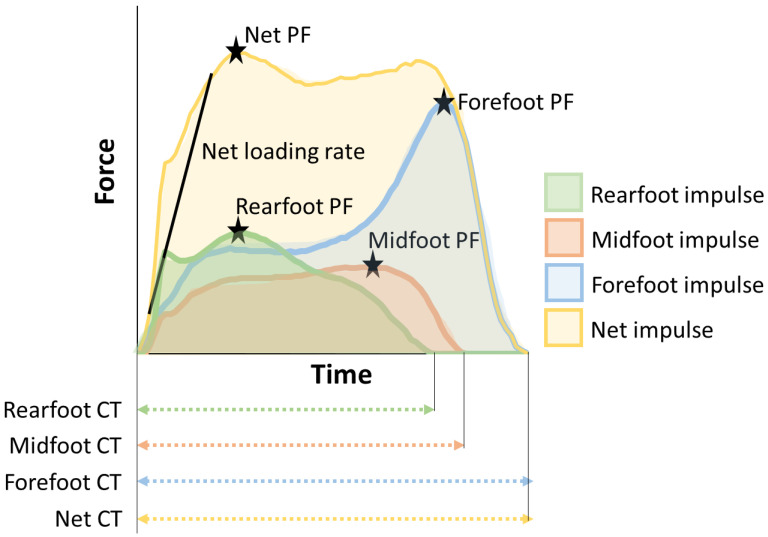
Example force–time data for each plantar region in one limb, collected during walking.

**Figure 4 sensors-24-05054-f004:**
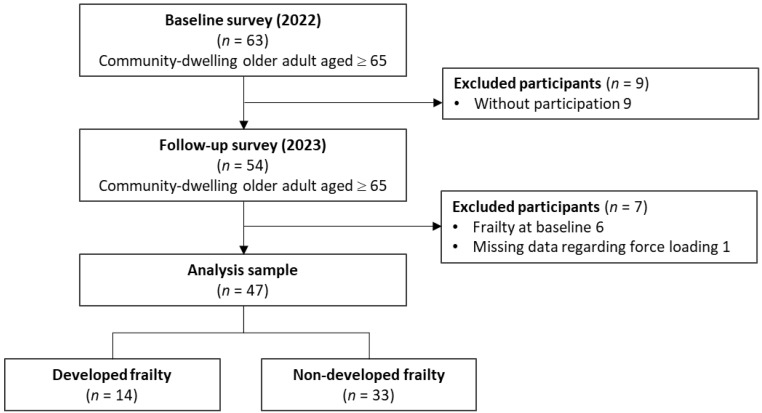
Flow chart of the participant recruitment process.

**Figure 5 sensors-24-05054-f005:**
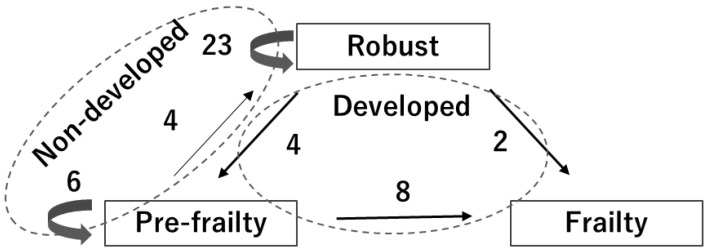
Six participants who were robust at baseline were reclassified into the pre-frailty or frailty groups during the follow-up period. Eight participants who were pre-frailty at baseline were reclassified into the frailty group during the follow-up period. These 14 participants were defined as a frailty development group. The remaining 33 participants who maintained robust or pre-frailty during the follow-up period were defined as the non-frailty development group.

**Figure 6 sensors-24-05054-f006:**
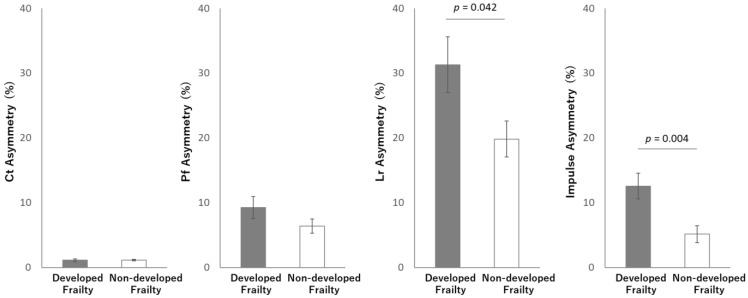
Measures of load asymmetry on net plantar force in subjects who were reclassified into the frailty group and those who were not during the 10-month follow-up period. For Lr and Impulse measures, asymmetry was increased significantly in those subjects who had become frail. Shown are the group means adjusted for frailty condition. The error bars reflect the standard error of the means.

**Table 1 sensors-24-05054-t001:** Baseline variables in groups which did or non-developed frailty.

	Developed Frailty	Non-Developed Frailty	*p*
N	14	33	
Age (y)	76.5 ± 6.7	74.7 ± 5.4	0.325
Women (%)	85.7	75.8	0.366
BMI (m^2^/kg)	22.9 ± 3.2	22.5 ± 3.1	0.664
KCL score	3.8 ± 1.6	2.6 ± 1.6	0.015
Fall history (%)	28.6	27.3	0.596
Cerebrovascular disorder (%)	3.0	0	0.702
Hypertension (%)	42.8	42.4	0.613
Heart disease (%)	14.2	3.0	0.208
Diabetes (%)	14.2	9.1	0.427
Liver disease (%)	0	0	
Kidney disease (%)	0	0	
Lung disease or/and asthma (%)	0	0	
Knee or/and foot pain (%)	27.3	21.4	0.489
Number of medications	2 ± 1	2 ± 1	0.852
Gait speed (m/s)	1.47 ± 0.23	1.44 ± 0.16	0.620
TUG time (s)	6.4 ± 0.9	6.5 ± 1.2	0.745

Abbreviations: KCL, Kihon Checklist; TUG, Timed up and Go.

**Table 2 sensors-24-05054-t002:** Multivariate logistic analysis of predictors of frailty development.

Variables	Adjusted OR	95% CI	*p*
Net impulse asymmetry (per 1-point increase)	1.269	1.016–1.585	0.036
KCL score (per 1-point increase)	2.331	1.170–4.647	0.016

Abbreviation: KCL, Kihon Checklist.

**Table 3 sensors-24-05054-t003:** Correlations in net impulse asymmetry.

	Sub-Regions	*r*	*p*
Contact time asymmetry	Forefoot	0.036	0.826
	Midfoot	−0.004	0.979
	Rearfoot	0.026	0.874
Peak force asymmetry	Forefoot	0.302	0.062
	Midfoot	0.323	0.045 *
	Rearfoot	0.123	0.457
Impulse asymmetry	Forefoot	0.246	0.126
	Midfoot	0.260	0.105
	Rearfoot	0.263	0.106

The r value is the Pearson correlation coefficient. * *p* < 0.05.

## Data Availability

The original contributions presented in the study are included in the article; further inquiries can be directed to the corresponding author.

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
