# Peer review of "In-Shoe Sensor Measures of Loading Asymmetry during Gait as a Predictor of Frailty Development in Community-Dwelling Older Adults"

_sensors, 2024, doi:10.3390/s24155054_

Round 1

Reviewer 1 Report

Comments and Suggestions for Authors

The paper calls: "In-shoe Sensor Measures of Loading Asymmetry during Gait as a Predictor of Frailty Development in Community-Dwelling Older Adults"  and concerned of method of measures of loading asymmetry by array of sensors. The main problem of article is absence of physical principal of operation of sensor. Is it FBG sensor, tension sensor or etc? How this sensor works?

Additional comments 

The primary question addressed by this research is whether increased limb loading asymmetry during gait, as measured by in-shoe sensors, can predict the development of frailty in community-dwelling older adults. 

The authors should consider including a larger sample size to improve the generalizability of the results. Additionally, incorporating a control group of younger adults or non-frail older adults might provide a clearer comparison. Ensuring consistency in the walking test conditions (e.g., footwear, walking surface) and controlling for confounding variables such as comorbidities and medication use would strengthen the methodology. The references appear to be appropriate, as they likely include previous studies on frailty assessment, gait analysis, and the use of wearable technology in biomechanical research. However, a detailed review of the reference list would be necessary to confirm their relevance and comprehensiveness.

end comment.

Reviewer 2 Report

Comments and Suggestions for Authors

The article presents the results of a study of limb loading asymmetry with the aim to predict frailty risk in community-living older adults. The authors performed a vast amount of research, and the resulting findings are undoubtedly valuable for the medical area of expertise. However, since the scope of the journal is sensor technology, the article lacks a comprehensive description of the technical aspects of the sensors used in the study, such as the type(s) of sensors, their operating principle and the structure of data acquisition system. The authors should highlight the novelty of the current work in terms of technical solutions used in the study. Otherwise, it is recommended to consider resubmission of the article to another journal with a corresponding scope related to medical research.

Reviewer 3 Report

Comments and Suggestions for Authors

Title: In-Shoe Sensor Measures of Loading Asymmetry during Gait 2 as a Predictor of Frailty Development in Community-Dwelling 3 Older Adults.

This manuscript needs in detail modification to increase its scientific value. 

Overall Comment : 

1. Where is the full proposed method diagram?

2. Please include all mathematical explanations. 

3. I found there are no clear scientific findings. 

4. If possible please add force sensor details. There are no detailed explanations. The reader will not understand what and how force sensors work if you don't explain. 

5. In the introduction write some bullet points about your main contributions. 

6. In related work explain the main challenge and others co-work. 

Comments on the Quality of English Language

Moderate editing of English language required

Round 2

Reviewer 2 Report

Comments and Suggestions for Authors

The authors have responded to all the comments, and the article is now acceptable for publication. 

Reviewer 3 Report

Comments and Suggestions for Authors

Carefully check the full manuscript. 

Comments on the Quality of English Language

No problem